# Implementing online consultations in primary care: a mixed-method evaluation extending normalisation process theory through service co-production

Michelle Farr,[1,2] Jonathan Banks,[1,2] Hannah B Edwards,[1,2] Kate Northstone,[1,2] Elly Bernard,[3] Chris Salisbury,[1,2] Jeremy Horwood[1,2]

[1]Population Health Sciences, Bristol Medical School, University of Bristol, Bristol, UK
[2]National Institute for Health Research Collaboration for Leadership in Applied Health Research and Care (NIHR CLAHRC) West, University Hospitals Bristol NHS Foundation Trust, Bristol, UK
[3]OneCare (BNSSG) Ltd, Bristol, UK

**Correspondence to**
Dr Michelle Farr;
m.farr@bristol.ac.uk

## ABSTRACT

**Objectives** To examine patient and staff views, experiences and acceptability of a UK primary care online consultation system and ask how the system and its implementation may be improved.

**Design** Mixed-method evaluation of a primary care e-consultation system.

**Setting** Primary care practices in South West England.

**Methods** Qualitative interviews with 23 practice staff in six practices. Patient survey data for 756 e-consultations from 36 practices, with free-text survey comments from 512 patients, were analysed thematically. Anonymised patients' records were abstracted for 485 e-consultations from eight practices, including consultation types and outcomes. Descriptive statistics were used to analyse quantitative data. Analysis of implementation and the usage of the e-consultation system were informed by: (1) normalisation process theory, (2) a framework that illustrates how e-consultations were co-produced and (3) patients' and staff touchpoints.

**Results** We found different expectations between patients and staff on how to use e-consultations 'appropriately'. While some patients used the system to try and save time for themselves and their general practitioners (GPs), some used e-consultations when they could not get a timely face-to-face appointment. Most e-consultations resulted in either follow-on phone (32%) or face-to-face appointments (38%) and GPs felt that this duplicated their workload. Patient satisfaction of the system was high, but a minority were dissatisfied with practice communication about their e-consultation.

**Conclusions** Where both patients and staff interact with technology, it is in effect 'co-implemented'. How patients used e-consultations impacted on practice staff's experiences and appraisal of the system. Overall, the e-consultation system studied could improve access for some patients, but in its current form, it was not perceived by practices as creating sufficient efficiencies to warrant financial investment. We illustrate how this e-consultation system and its implementation can be improved, through mapping the co-production of e-consultations through touchpoints.

### Strengths and limitations of this study

► Largest UK study to date examining staff and patient experiences of using a primary care online consultation system.
► Extending normalisation process theory with service co-production theory enables an in-depth understanding of how patients and staff interacted with the e-consultation system.
► Touchpoint analysis enables improvements to be suggested to develop the design and implementation of online consultation systems, aimed at software designers, policy-makers and general practices interested in this technology.
► This observational study was based on a pilot period of one online consultation system, and issues highlighted may be due to the system studied, rather than all online consultation systems.

## BACKGROUND

English general practice clinical workload has risen by 16% over the period 2007–2014.[1] Ninety three per cent of general practitioners (GPs) say their workload has negatively impacted on quality of care given to patients.[2] Average waiting times for an appointment have been reported as just under 13 days.[3] Internationally, policy-makers are advocating technological alternatives to face-to-face primary care consultations to improve service quality.[4] UK policy to improve primary care access includes the Prime Minister's Challenge Fund (now the GP Access Fund),[5] and the General Practice Forward View.[6] These promote greater use of technology to manage workload and improve patient access, with £45 million made available to support the implementation of online consultation systems.[7] Online or e-consultations enable patients to contact their GP via a mobile app

| NPT Construct | |
|---|---|
| **Coherence** | Sense-making work to understand the possibilities of an intervention. *What are the purposes of e-consultations?* |
| **Cognitive participation** | Relational work that builds a community of practice around an intervention. *What promotes participation with e-consultations?* |
| **Collective action** | Operational work that people enact to make an intervention function. *How do participants interact with e-consultations to make them work?* |
| **Reflexive monitoring** | Appraisal work where people assess how a new practice affects them and others. *How do participants appraise e-consultations?* |

**Figure 1** Normalisation process theory (NPT) constructs in association with the implementation of e-consultations.

or online portal.[7] General practice staff attitudes toward electronically based consultations include concerns about medicolegal issues, clinical limitations and increasing workload concerns.[8–10] Research into practitioners' and patients' actual experiences of e-consultations is limited, but timely, as implementation is at an early stage.[4 10]

A consortium of general practices in South West England (One Care),[11] received funding through the GP Access Fund,[5] to pilot online consultations. Starting in April 2015, the eConsult system[12] (previously called WebGP) was implemented free of charge into 36 general practices. The eConsult system was designed by GPs, software programmers and operational managers, with support from medical defence organisations.[13] Patients access the eConsult system (referred to as 'the system' in this paper) via their own GP practice website. They can access self-help, pharmacy advice, 111 (National Health Service (NHS) non-emergency telephone advice), administrative help (such as repeat prescriptions) or submit an online form with details of their condition, electronically sending this to their GP practice, where it is then processed. If the system identifies signs or symptoms that may require immediate medical attention, patients are redirected to appropriate services; otherwise, the system informs patients that their GP practice will contact them by the end of the next working day.

Normalisation process theory (NPT) illustrates issues to address when implementing a technology or complex intervention (figure 1).[14–16]

Patients' perspectives of implementing technology have been researched less,[17] and NPT may need to be developed to account for patients' implementation roles.[18–21] With e-consultations, patients input details of their symptoms, which produces the e-consultation that the practice then processes. In this way, an e-consultation is co-produced; both patients and staff are integral to the process. This article examines co-production 'in the implementation of core services' where 'citizens are actively engaged in the implementation, but not the design, of an individual service' (Brandsen et al, p433).[22] We develop NPT to analyse

patients' implementation roles, using service co-production theory[22–29] to understand how both patients and staff co-implement and use technology.

We undertook an evaluation of eConsult to analyse patient usage, acceptability, effectiveness and costs of implementing the system in the 36 general practices, incorporating a quantitative, qualitative and economic analysis. The quantitative and economic analysis on usage and costs[30] and interviews with practice staff about e-consultations[31] are reported separately. This article analyses the implementation and acceptability of the eConsult system from patient and staff perspectives, using NPT[14] and service co-production theory[22–29] to understand their experiences and how the e-consultation system and its implementation may be improved.

## METHODS
### Research design
Data were collected that covered up to 15-month usage of the system by GP practices, and consisted of three components:
1. qualitative interviews with staff from a sample of six GP practices,
2. quantitative data from electronic medical records for patients who had conducted an e-consultation from a sample of eight GP practices,
3. quantitative and qualitative patient survey data from patients who had conducted an e-consultation about their experiences of e-consultations from all 36 GP practices.

### Sampling and recruitment
To conduct qualitative staff interviews and collect anonymised patient record data, GP practices were purposively sampled to ensure a range of: locations (rural/suburban/urban); practice levels of deprivation measured by the Index of Multiple Deprivation from practice postcodes and volume of e-consultation usage (calculated by dividing the number of e-consultations received by the number of

**Table 1** Sampled GP practice and interview participant profiles

| GP practice | e-Consultations per day live (range 0.1–2.9 for 36 practices) | IMD deciles of deprivation[51] (lower decile=more deprived) | Area | Ethnic minority population (%) | Staff interviews | Number of e-consultations randomly sampled from electronic patient record data |
|---|---|---|---|---|---|---|
| 1 | 2.9 | 5 | Urban | 17.5–20 | GPs 2, AD 1, PM 1 | 64 |
| 2 | 0.9 | 10 | Rural | 0–2.5 | GPs 2, AD 2, PM 1 | 60 |
| 3 | 1.6 | 1 | Urban | 35–37.5 | GPs 2, AD 1, PM 1 | 70 |
| 4 | 0.2 | 1 | Urban | 7.5–10 | GP 1, AD 1, PM 1 | 0 |
| 5 | 0.7 | 3 | Urban | 5–7.5 | GP 1, PM 1 | 38 |
| 6 | 0.8 | 8 | Urban | 10–12.5 | GPs 2, NP 1, AD 1, PM 1 | 0 |
| 7 | 2.2 | 5 | Urban | 12.5–15 | 0 | 60 |
| 8 | 1.2 | 9 | Suburban | 7.5–10 | 0 | 60 |
| 9 | 0.6 | 10 | Urban | 7.5–10 | 0 | 66 |
| 10 | 1.5 | 9 | Urban | 7.5–10 | 0 | 67 |

AD, administrator; GP, general practitioner; IMD, Index of Multiple Deprivation; NP, nurse practitioner; PM, practice manager.

days the system was live at time of sampling). Table 1 illustrates the range of practices recruited, with details of the eight practices purposively sampled to collect anonymised patient medical record data from e-consultations, and the six practices purposively sampled to conduct qualitative interviews. A purposive sample of staff with different professional roles from these six practices involved in the processing or managing of e-consultations were invited to be interviewed via email with participant information sheets, with contacts and invitations facilitated by practice managers (PMs). Practice protocols on using the system were also given to the researchers where they were available.

## Qualitative interviews and analysis

Interviews were conducted with 23 practice staff including: 10 GPs, 1 nurse practitioner (this interviewee is designated with (GP) notation to avoid potential identification), 6 practice managers (PM), and 6 'administrators' including an information technology manager and receptionists (administrator) (see table 1). Interviews took place both face to face within general practice private offices (n=20) and over the phone (n=3), and lasted between 10 and 40 min. All participants gave full informed consent. Interviews were semistructured, using a topic guide that had been developed with reference to NPT[14] covering: (1) introduction of e-consultations into the practice; views, promotion, training needed, (2) e-consultation processing, (3) impact on workload, (4) impact on clinical practice and (5) attitudes to future implementation. All interviews were audio recorded, transcribed, anonymised, checked for accuracy and imported into NVivo 10 software to aid analysis. Inductive thematic analysis was used grounded in the data[32]; NPT was then used as a framework to order the codes. Analysis was conducted by two researchers (MF and JB) with a subset double coded to ensure rigour. Data collection

and analysis were conducted in parallel, with participants sampled until data saturation was reached. Key analytic themes were discussed within the research team to enhance credibility and external validity.

## Patient record data

Electronic anonymised patient record data were collected from a random sample of patients (n=485) who had used e-consultations from eight of the participating practices (table 1), between April 2015 to June 2016. A staff member from a participating practice retrospectively extracted anonymised patient data from patient records onto an Excel database from all practices, including: patient demographics; reason for contact; the actions taken resulting from the e-consultation (eg, telephone call, face-to-face appointment, email advice) and further care provided by the practice in the next 30 days in relation to the e-consultation (eg, treatment room tests, nurse appointments, further GP consultations, etc). When analysing practice responses to e-consultations, the primary response was designated as the most resource-intensive action (ie, a face-to-face appointment is more resource intensive than a phone appointment than a prescription), and the secondary action was the next most resource intensive (ie, a prescription or advice) to account for multiple e-consultation processes.[30] The primary clinical reason for patients using an e-consultation was cross tabulated with the primary response to the e-consultation from practice staff using descriptive statistics.

## Patient survey data

Patient survey data were routinely collated by the e-consultation software company (Hurley), using their own survey design. Patients who opted in were sent a questionnaire 7 days after the submission of their e-consultation. This contained both tick box questions and free text. We were given access to this anonymised data from the software

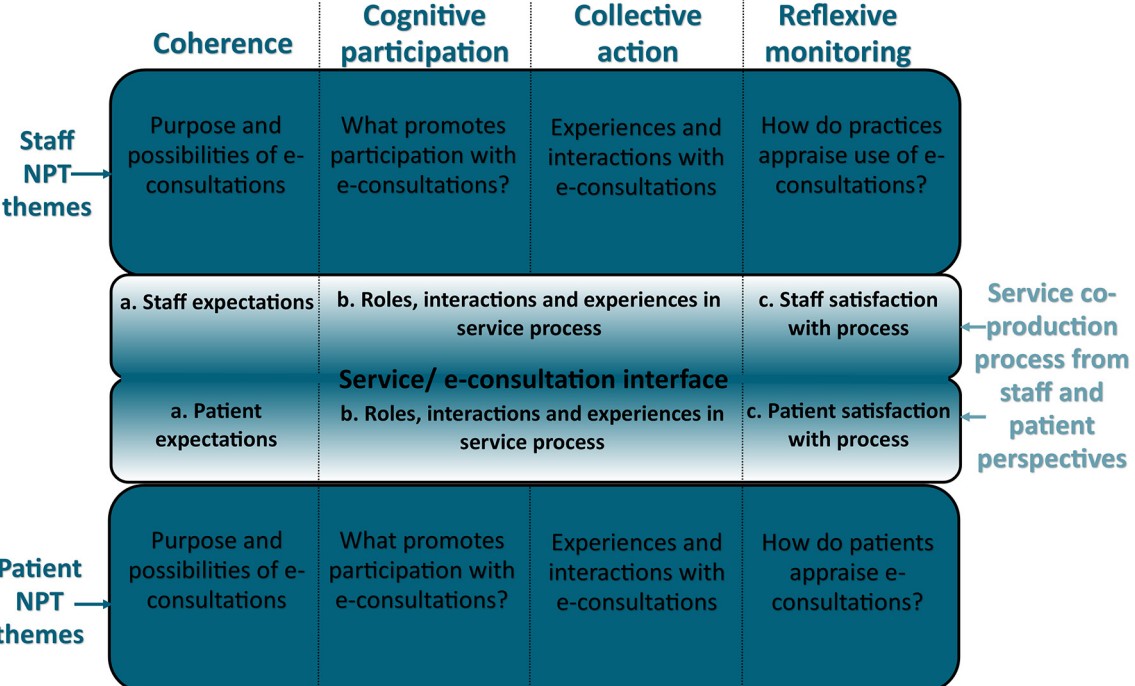

**Figure 2** Combining normalisation process theory (NPT) framework with service co-production processes.

company for the 36 pilot GP practices from April 2015 to June 2016. The tick box questions were analysed using descriptive statistics (online supplementary file, table A). Free-text comments were coded using inductive thematic analysis grounded in the data[32]; NPT was then used as a framework to order the codes. Answers were analysed by two researchers (MF and JB), with a subset double coded. Patient survey quotes are labelled P01, P02… in the following data analysis. From 7472 e-consultations, a total of 751 patients (10%) submitted a survey with quantitative data, and additional comments to individual questions ranged from 38 to 512 patients (online supplementary file, table A,B). Qualitative patient survey data were used to facilitate interpretation of the quantitative patient survey responses.

### Theoretical integration of patient and staff data using NPT and co-production theory

Service co-production theory and NPT were theoretically integrated to examine implementation from staff and patient's points of view (figure 2). This also enabled us to examine the processes and interactions between patients and staff when using the e-consultation system. Service co-production can be understood as a process where service quality is shaped by (1) people's initial *expectations* of a service, (2) staff and service users' *roles, interactions and experiences* within a service, leading to (3) their *resulting satisfaction or dissatisfaction*.[23 29 33] Understanding this process helps to analyse service users' roles as a co-producer of a service.[26 34] NPT constructs[14 16] and service co-production processes[23] can be integrated together and used to analyse staff and patients' initial expectations, interactions with and experiences of e-consultations, and

their subsequent perceptions resulting in satisfaction/dissatisfaction (figure 2).

Patient survey data (quantitative and qualitative), staff interview data and patient record data were theoretically integrated,[35] bringing different findings together into this theoretically informed framework (figure 2). The NPT concepts of *coherence* and *cognitive participation* were analysed using staff interviews and patients' survey responses. *Coherence* explored staff and patients' expectations of the system and how the system's purpose and possibilities were understood. *Cognitive participation* explored the relational work that promoted engagement with e-consultations. *Collective action* explored how the system was operationalised. Initially, an e-consultation workflow process map for each practice was developed from staff interviews and practice protocols on using the system. These were integrated to illustrate core practice processes. 'Touchpoints' (points of contact and interaction through a service process) were identified by using service blueprint techniques to map the e-consultation process.[25 36] Service blueprints are maps of service systems that illustrate service user and staff roles, actions and interactions, and can illustrate how service users expectations and experiences affects service quality.[25 36] Using staff interviews of the e-consultation process, and qualitative patient survey responses, three touchpoints[25 34] were identified, where patients and staff interacted through the e-consultation process. Touchpoints have been used in co-production literature[25 34] and health service improvement methods such as experience-based co-design.[37–39] Here, they were used as an analytic lens

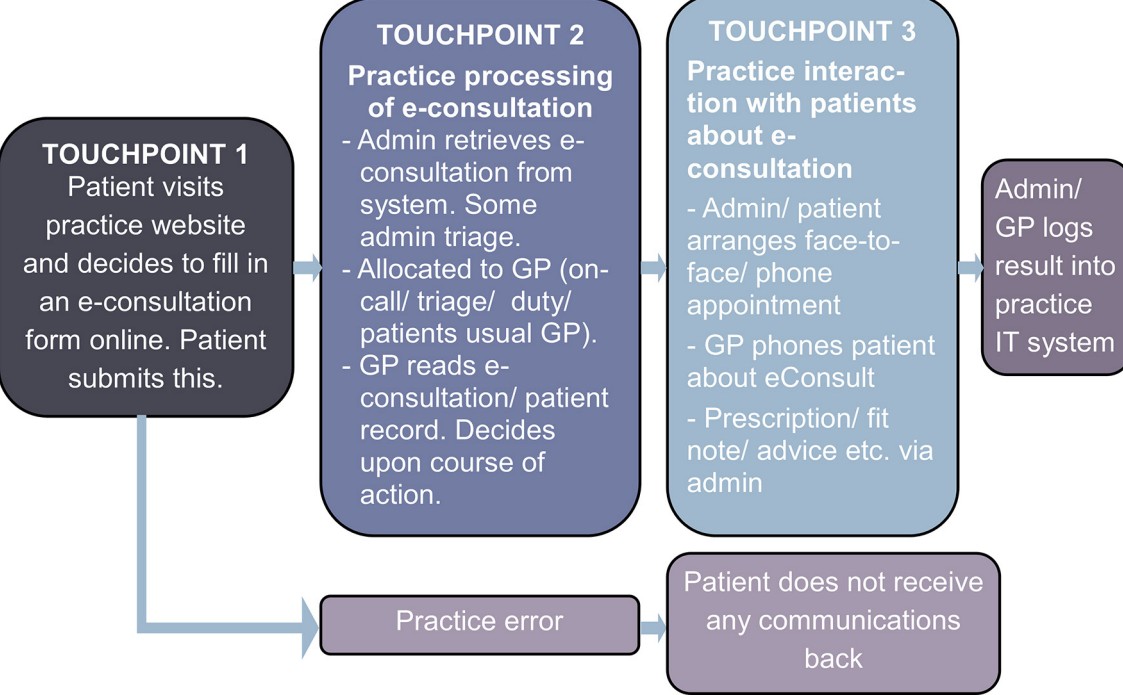

**Figure 3** e-Consultation process map highlighting key touchpoints. GP, general practitioner.

to examine the operational work and experiences of both staff and patients through an e-consultation. Key touchpoints are illustrated in figure 3 and analysed in the *collective action* results section.

*Reflexive monitoring* explored staff and patient appraisal of the system, analysing when e-consultations may work for whom. Patients' clinical reasons for using an e-consultation and practice staff responses from patient record data (table 2) were integrated

with the analysis of qualitative staff and patients' comments about their satisfaction with the system, integrating all data sets. This integration of qualitative and quantitative data used established 'following a thread'[35] techniques where the question of why staff and patients were satisfied/dissatisfied with the system was traced using all data sets, to understand patients and staff sources of satisfaction/dissatisfaction with the system.

**Table 2** Primary response from practice staff by reason for e-consultation (from patient record data)

| Patient reason for consulting | GP practice staff response to e-consultation | | | | | | | |
|---|---|---|---|---|---|---|---|---|
| | Total number (%) | Face to face % | Phone consult % | Prescription % | Fit note % | Test request % | Refer routine % | Advice % | Other/ unknown % |
| Musculoskeletal/limb pain | 60 (12.4) | 48.3 | 38.3 | 1.7 | 0 | 1.7 | 3.3 | 1.7 | 0 |
| Infection/immunological | 70 (14.4) | 40.0 | 41.4 | 8.6 | 0 | 0 | 0 | 0 | 0 |
| Neurological | 26 (5.4) | 53.9 | 26.9 | 0 | 0 | 3.9 | 0 | 0 | 3.9 |
| Sexual/reproductive health | 41 (8.5) | 39.0 | 41.5 | 7.3 | 0 | 4.9 | 0 | 0 | 2.4 |
| Dermatological | 33 (6.8) | 48.5 | 21.2 | 18.2 | 0 | 0 | 0 | 3.0 | 0 |
| Respiratory | 25 (5.1) | 52.0 | 24.0 | 4.0 | 0 | 0 | 0 | 0 | 8.0 |
| Mental health | 29 (5.9) | 44.8 | 34.5 | 10.3 | 0 | 0 | 0 | 0 | 0 |
| Digestive | 19 (3.9) | 52.6 | 26.3 | 5.3 | 0 | 0 | 0 | 0 | 5.3 |
| Medication query/advice | 19 (3.9) | 0 | 73.7 | 10.5 | 0 | 0 | 0 | 0 | 5.3 |
| Administrative* | 109 (22.5) | 12.2 | 27.1 | 11.2 | 14.0 | 1.9 | 5.6 | 10.3 | 7.5 |
| Other/unclear | 54 (11.1) | 38.4 | 17.0 | 0 | 0 | 3.8 | 0 | 5.7 | 1.9 |
| **Total** | **485 (100)** | **38.1** | **32.1** | **7.2** | **3.1** | **1.6** | **1.6** | **9.1** | **6.4** |

*Fit notes, test results, referrals, repeat scripts, letter requests and booking appointments.
GP, general practitioner.
Bold values, total number of e-consultation patient records and percentage of responses to an e-consultation.

## RESULTS

The results are presented using the four NPT concepts, as detailed above.

## Coherence

Coherence describes patients' and staff understandings and expectations of the system's purpose. e-Consultations were seen by practice staff as a new way of working that had the potential to reduce GP workload pressures:

> We are massively overstretched … So, that was one of the reasons why I wanted [eConsult], was so that we could make it easier… to deal with queries and often relatively simple queries that come through (PM23).

Practices were aware of the difficulties patients faced in securing GP appointments and e-consultations were seen to provide a different pathway to care and advice. The pilot provided PMs with an opportunity to test out the system without financial investment.

Patients saw e-consultations as a new, alternative way to communicate with their practice, that could be used out of surgery hours, *'It is quick and easy to use at a time to suit myself. Saves having to call the surgery'* (P61). Several patients' comments exhibited an understanding of the pressures that practices were under: '*It saves the GP time, saves me coming to the practice, great all round'* (P81).

## Cognitive participation

Cognitive participation describes the relational work that people were involved in to promote participation with the system. Implementing e-consultations within practices was reported by practice staff as a relatively straightforward process, with little training needed. However, there were varying feelings towards it:

> We were quite happy to do it (AD08).

> I was feeling very anxious about the extra workload … some things feel like a bottomless pit (GP22).

Few practices reported involving patients in implementing e-consultations; one practice mentioned that their patient participation group was concerned that the system may disadvantage those who were less able to use technology. Practices employed different promotion methods to patients to varying degrees including through their website, waiting room banners, leaflets, social media, on phone answering messages and newsletters. In some practices, there was a feeling that there was not as much uptake of the system as expected.

Some patients were activated to use the system because they could not get an appointment: *'No available appointment for 2 weeks'* (P10); *'Tried Switchboard nine times … Went online'* (P05). Others favoured the online format and remote consultation style; they used the system as it was difficult to visit the practice due to disabilities, illness or working commitments, or saw it was a more legitimate way to access GP advice: *'didn't want to waste Drs time with a full consultation which I didn't need'* (P171).

## Collective action

*Collective action* describes how the system was operationalised in practice by patients and staff. figure 3 maps where staff and patients interacted through an e-consultation process, identifying three touchpoints, key interactions and experiences through the co-production of an e-consultation.

### Touchpoint 1: patient interaction with e-consultation system

Touchpoint 1 in figure 3 relates to patients' initial decision to complete an e-consultation form, and their interaction with the system. Over the 15-month pilot period, 7472 patients completed an 'e-consultation', most frequently on weekdays and during traditional working hours.[30] Patient record data show that women used e-consultations more than men (64.7% vs 35.3%) and 53.4% were between 25 and 44 years old.[30] Most commonly, patients submitted administrative requests, for example, repeat prescriptions, test results and letters (22.5%), followed by immunological/infection issues (14.4%) (see table 2 and Edwards *et al*.[30]). Most patient survey respondents agreed that the system was easy to use (online supplementary file, table A): *'had no problems at all'* (P398). It was '*helpful to be able to contact about minor requests'* (P475). Some patients preferred the written interface over a verbal conversation: '*Allowed me time to describe symptoms in greater detail than talking'* (P279). However, patients reported that the system did not seem to account for multiple conditions.

### Touchpoint 2: GP practice processing of e-consultations

During the pilot, the system was not integrated with the patient record IT system that practices used. Administrators downloaded patients' e-consultations from the system and printed them or manually imported them into patients' records. Some administrators spoke of conducting some triage, for example, directing hay fever queries to pharmacy. Clinicians described variability in the quality of information from the e-consultation forms. While information could be: '*clear and concise'* (GP13), this was not always the case:

> One patient needed to be admitted [to hospital]… Because the symptoms weren't very clear (GP05).

Most GPs often reverted to face to face or phone conversations to gain more information to conduct clinical decision-making.[30 31] One clinician, who had substantial experience of conducting phone triage, reported that they dealt with most e-consultations without needing to see patients face-to-face, unless it was for new acute symptoms/diagnosis.

### Touchpoint 3: GP practice interaction with patients following their e-consultation

Practices organised follow-up appointments in different ways. In some, a face-to-face appointment might have to *'start from scratch'* (GP05), because a different clinician originally dealt with the e-consultation:

I had to repeat everything I entered on line. What's the point in asking if you're not going to read it? (P90)

Other practices had more continuity where GPs could follow through the e-consultation, which provided benefits to the consultation:

The actual face to face consultation is then quicker, and that's quite nice in some ways …it doesn't open up other avenues, to a degree, okay, so it's more efficient (GP18)

Reception staff usually contacted patients via a practice email address or phone, to relay a message from a GP to patients, or to arrange the next step or outcome of their e-consultation. Occasionally, patients who had had no opportunity to speak to a doctor, were unhappy about this: '*I had no opportunity to ask any questions*' (P44). Some patients reported missing practice phone calls, one spoke of '*telephone answer machine ping-pong*' (P275). 14% of survey respondents reported not being contacted at all following their e-consultation (surveys were sent 7 days after an e-consultation submission) which left patients dissatisfied (online supplementary file, table A):

I feel like my treatment has been compromised and delayed as a result of this service (P48)

The system had an in-built function to electronically respond to a patient's email address; however, only one out of six practices where interviews were conducted said they used this, and not all staff could access the system.

Table 2 cross tabulates the primary clinical reason for patients using an e-consultation with the primary response to the e-consultation from practice staff. GP responses varied with patients' health queries, for example, medication queries and advice resulted in no face-to-face appointments, while 54% of neurological queries resulted in face-to-face appointments.

Patient record data in table 2 illustrate that overall 38% of e-consultations resulted in a primary response of a face-to-face consultation.[30] Several patients commented that they had received easier access to a face-to-face appointment through the system:

The service recommended immediate attention that resulted in a quicker appointment than otherwise would have been the case (P313)

While a face-to-face consultation often satisfied patients, it could potentially duplicate GP workload,[31] with initial e-consultation processing by administrators and a GP, plus an appointment space. 32% of e-consultations resulted in a primary response of a phone consultation. Where e-consultations resulted in a primary response of a prescription (7.2%), a 'fit note' statement of fitness for work (3.1%), test or treatment request (1.6%), referral (1.6%) or advice (9.1%) (occurring in 23% of patient e-consultation records),[30] these could save GP time as administrative staff relayed

messages and there was no direct contact between the patient and GP.

It helps in terms of administratively if there are things which can be done very simply, and that can free up, that can free up surgery time, to a degree (GP18).

### Reflexive monitoring: Who do e-consultations work for, and when?

*Reflexive monitoring* describes how patients and staff appraised the system and their resulting satisfaction/dissatisfaction. All data sets are integrated to explore when e-consultations were likely to work best for whom.

Patients' satisfaction with the system was high and most (81%) were likely to recommend the system to others. 76% said they would use the service again instead of booking a face-to-face appointment (online supplementary file, table A). Dissatisfaction with the system was usually a result of: lack of interaction with a GP; missed communications; thinking that their query could be answered remotely, and then being asked to book an appointment; or lack of timely follow-up of their e-consultation. Several patients suggested improvements (at touchpoint 1) that have since been integrated into the system by the software developers, for example, allowing patients to consult with multiple symptoms for both new and existing conditions; the ability to upload photographs; being able to nominate a preferred GP; simplification of language[40] and an administration channel for requests such as a fit note or test results.

Interviews revealed that clinical staff were less satisfied with the system, as time saved in completing e-consultations without further GP–patient communication (23%), was counterbalanced by e-consultation processing and GPs needing to phone or see patients in 70% of e-consultations, which could duplicate GP work.[31] Analysing why patients were e-consulting and the resulting action (table 2), and combining this with staff and patients' appraisal of the system, table 3 summarises when e-consultations were likely to work and be effective for patients and GPs. For GPs, it was only for relatively straightforward queries that the system could save substantial time. Patients were satisfied more often as e-consultations could: save them time, get them a quicker appointment, provide an easier access route to GP services or they preferred the remote access format.

Practice suggestions for system improvement (at touchpoint 1) included that patients could be signposted away from consulting a GP more often, to encourage more self-help or use of pharmacy when '*appropriate*' (AD11), to '*make people aware that they're in some cases wasting GP's time*' (AD04). Some practice staff suggested that patients might be guided to use e-consultations under certain conditions where only remote GP input was likely to be needed (as in table 3). In contrast, if a patient had multiple symptoms for a new condition for which a face-to-face appointment was likely to be needed, GPs suggested that a modified system could flag this, directing patients not to submit an

**Table 3** Nature of e-consultations and the resulting possible satisfaction and dissatisfaction of staff and patients

| Nature of query | Patients' satisfaction | Practice staff satisfaction |
|---|---|---|
| Administrative queries | ✓<br>Most processed remotely | ✓<br>Most processed remotely |
| Medication queries and simple queries about pre-existing patient conditions | ✓<br>Most processed remotely by phone or prescription | ✓<br>Most processed remotely by phone or prescription |
| Queries about new conditions | ✓<br>May get quicker response | X<br>Face-to-face appointment more likely—possible work duplication |
| Complex questions, multiple symptoms | ✓<br>May get quicker response | X<br>Face-to-face appointment more likely—possible work duplication |

e-consultation but to directly book a face-to-face appointment, to avoid GP work duplication. Some practice staff were also concerned that the system might exacerbate inequalities of access for people with literacy difficulties or whose first language is not English, and those with difficulties in using a computer or mobile device.

Comparatively analysing different practice processing of e-consultations (touchpoint 2) suggests that administrative allocation of e-consultations to GPs could affect process efficiency. If administrators allocated e-consultations to a GP who had previously seen the patient (especially about similar symptoms/conditions) this may support more efficient processing, as GPs would be more familiar with the patient and condition:

> We like to look at each patient's notes to find out which doctor perhaps has seen this patient for that particular problem and then we would know where to direct that e-consultation (AD09)

Administrators could also book face to face or phone consultations with the GP who had processed the e-consultation and was familiar with the patient query. This could focus the appointment, and avoid situations where patients felt that GPs appeared not to have read their e-consultation.

Improvements at touchpoint 3 (practice interaction with patients about e-consultations) include more robust practice communication mechanisms to reduce patient dissatisfaction about practice communication relating to their e-consultation. This could be supported by integration with electronic practice IT systems,[17] and further use of electronic communications back to patients that more staff can access and use.

Summarising this touchpoint analysis highlights potential improvements to the system and its implementation (table 4).

None of the 36 practices took up the system after the pilot, which would have involved paying market prices for the software. However, 13 practices were interested in continuing to use the system if costs were paid for by alternative funding sources, and technological interoperability with electronic patient record systems was further developed.

## DISCUSSION
### Key findings
Practices were originally interested in the system to improve access and create efficiencies. While some patients used the system to try and save time for both themselves and their GPs, other patients were activated to use e-consultations when they could not get a timely appointment. Because practices were dependent on patients deciding how and when to use e-consultations, clearer guidance may be needed for patients to support more efficient use of e-consultations (see table 4, touchpoint 1).

Our findings highlight the difficulties in substituting real-time interaction with an asynchronous technological interface (touchpoints 2 and 3). This could reduce professionals' ability to use tacit knowledge of patients concerns, patients' ability to negotiate treatment options and shared decision-making. GPs often needed further information when processing e-consultations, leading to face-to-face and phone consultations, which could duplicate workload. However, the system was being piloted, which meant that GPs were developing their skills in e-consultations, so phone and face-to-face consultations may decrease over time. GPs speculative fears about the perceived risks to patients of online consultations and the potential increases in workload[2 10] are to some extent causally linked through this study. For more efficient implementation of e-consultations, further consideration may be needed of when it is appropriate to use technology, for example, for less complex tasks,[9] and when face-to-face interaction is essential, such as in the diagnosis of complex symptoms.[41]

Other interventions designed to improve efficiency and access in primary care highlight potential workload issues; for example, nurse-led telephone triage may reduce GP contact time, but increase overall clinician contact time.[42] Previous e-health studies that use NPT highlight barriers

**Table 4** Suggested improvements to implement the e-consultation system

| Issues identified with touchpoint 1: patient decides to fill in an e-consultation form | Suggested technological improvements |
|---|---|
| Patients suggested several ways to improve system usability, such as: allowing patients to consult with multiple symptoms for both new and existing conditions; the ability to upload photographs; being able to nominate a preferred GP; simplifying language and an administration channel for requests such as a fit note or test results | Software developers have implemented these improvements to the system in its ongoing development[40] |
| Practice staff suggested that the system could encourage more use of pharmacy or self-help options where appropriate | Better signposting to pharmacy and self-help options on website interface |
| Promoting patients to use e-consultations for simple conditions and questions to save face-to-face appointments | Appropriate patient signposting on when to complete an e-consultation |
| Reducing patient e-consultation usage when they need a diagnosis about new, complex and multiple symptoms | Appropriate patient signposting on when not to complete an e-consultation but to directly book a face-to-face appointment to save practice staff work duplication |
| Reducing the use of the e-consultation system to directly access face-to-face appointments | Signposting to discourage patient use of the system if they want a face-to-face appointment |
| **Issues identified with touchpoint 2: practice processing of e-consultations** | **Suggested practice implementation improvements** |
| Some GPs received e-consultations that could have been dealt with by a pharmacy | Administrative triage where appropriate, for example, directing hay fever queries to pharmacy, to save GP time |
| Supporting more efficient processing of e-consultations, and potentially reduce follow-on face-to-face consultations | Allocate e-consultations to GPs who are familiar with the patient and their symptoms, where appropriate |
| **Issues identified with touchpoint 3: practice interaction with patients about e-consultations** | **Suggested improvements** |
| Patient complaints that they had to repeat information in consultation as GPs appeared not to have read e-consultations | Allocate follow-on phone and face-to-face appointments to GPs who initially process e-consultations |
| Patients missing or not receiving communication back from the practice about their e-consultation | More robust internal practice systems to ensure that patients receive communication back about their e-consultations. Stronger e-consultation and practice IT integration to support electronic communications back to patients that more practice staff can access and use |

GP, general practitioner.

of adverse effects on workload[43] and poor interactional workability of technology which can impede adoption within primary care.[44] e-Consultations supported efficiencies for straightforward GP queries, but less so complex ones, showing that how patients use technology can affect its implementation.[45] Our results align with other studies that highlight potential barriers to technological implementation including that: the clinical data the system was designed to generate from patients were sometimes incomplete[46]; the system was not fully interoperable with other IT systems, and costs prohibited long-term usage.[17]

### NPT and service co-production theory

Service co-production theory and touchpoints can extend NPT through focussing on how technologies change the service process and interactions between patients and staff. While involving patients voluntarily in co-designing technology may improve e-health technology[17 39]; this paper's contribution illustrates how service co-production theory can support the analysis of how patients co-implement technology through everyday service interactions, rather than voluntarily being involved in co-designing a service. Service co-production particularly extends the *collective action* aspects of NPT, exploring in depth how both staff and patients operationalise and relate through a service system. Touchpoint analysis illustrates how patients and staff responded to digital prompts and interacted through the e-consultation process. This fills a research gap to specifically examine how e-health services affect clinical interactions with patients.[43 45] It shows how e-health implementation may be reconfigured through staff and service user produced knowledge[43] to improve technology and its implementation. This may tackle barriers to technological adoption, such as understanding how technology impacts care delivery, relationships between caregivers and receivers, the role of patients in implementation and how to maintain and improve ongoing implementation.[17 45]

## Policy and practice implications

Technology is often promoted to improve NHS efficiency,[47] but benefits are often more limited due to implementation difficulties.[17] In this study, no practices experienced sufficient workload savings to warrant practices own financial investment in the system at current market prices; however, the system did improve access for some patient groups. NHS England has offered financial support for practices to adopt online consultations.[6] Our research affirms that clear implementation guidance is needed[48] and provides recommendations (table 4) to support the technological developments of e-consultations and future implementation to alleviate additional GP workload while improving patient access. NHS England case studies of e-consultation systems include their potential role to triage most patients.[49 50] While our study gave no statistical evidence that patient socioeconomic factors affected usage rates,[30] practitioners in our qualitative study had concerns about the system's potential impact on equality of access. Further research is needed to investigate equity of access when implementing e-consultations.

## Strengths and limitations

This study is one of the largest UK pilot independent evaluations of e-consultation systems to date, covering a wide range of GP practices. The broad sample of practice staff interviewed, combined with patient record data and patient survey data, allows a comprehensive insight into the e-consultation system. Patients' qualitative survey comments varied in depth, but provided a wide breadth of responses, for example, 510 respondents explained reasons for satisfaction/dissatisfaction with the system. An early internal research report shared with the e-consultation software developers has supported improvements to the e-consultation system studied.[40] Theoretically, combining NPT and co-production theory has enabled the integration of staff's and patients' perspectives, and touchpoint analysis has suggested further improvements that can be developed. However, because the study was based on a pilot period of one online consultation system, the issues highlighted may be a result of the system studied, rather than all online consultation systems. Patient surveys were only sent to patients who had submitted an e-consultation (of which 10% responded), thus representing a self-selecting sample of those who had invested time into the system. Surveys were sent to patients 7 days after they had submitted an e-consultation, which may have been before their e-consultations had been processed with 14% of patients waiting to hear back. Because e-consultation usage was low,[30] those patients using the system may be unrepresentative of the wider patient population.

## CONCLUSIONS

e-Consultations can increase patient access and satisfaction, but in their current form, were not perceived as creating sufficient workload efficiencies for continued practice usage. Patients' use of e-consultations impacted on staff's appraisal of the system. Where both patients and staff interact with healthcare technology, it is in effect 'co-implemented'. Extending NPT through service co-production theory and touchpoints enables an analytic focus on service processes and interactions between staff and patients, and how the e-consultation system affected these. Mapping the co-production of an e-consultation through touchpoints[34 36] has highlighted where the system may be redesigned or implementation improved. This analysis can support more effective implementation of appropriate technology that accounts for professional and patient experiences.

**Acknowledgements** Authors thank staff from participating practices for assisting with the collection of individual-level data, and the software developers for providing web usage statistics and anonymous patient survey data.

**Contributors** KN, JH, HBE, MF, JB and EB were responsible for the study design and collection of data. KN, JH and EB were responsible for study management and coordination. CS was a project advisor throughout. MF, JB, HBE and KN analysed the data. MF drafted the paper. All authors read, commented on and approved the final manuscript.

**Funding** This research is funded by the One Care Consortium Ltd and the National Institute for Health Research (NIHR) Collaboration for Leadership in Applied Health Research and Care West at University Hospitals Bristol NHS Foundation Trust. The One Care Consortium facilitated data collection.

**Disclaimer** The views expressed in this article are those of the authors and not necessarily those of the NHS, the NIHR, or the Department of Health and Social Care.

**Competing interests** None declared.

**Patient consent** Detail has been removed from this case description/these case descriptions to ensure anonymity. The editors and reviewers have seen the detailed information available and are satisfied that the information backs up the case the authors are making.

**Ethics approval** The study was reviewed by the NHS Health Research Authority (project ID: 204925) and ethically reviewed by the University of Bristol, Faculty of Health Sciences, Research Ethics Committee (Application 32961).

**Provenance and peer review** Not commissioned; externally peer reviewed.

**Data sharing statement** The datasets analysed during the current study are not publicly available, as participants were not asked to consent to this at the time of data collection. Related patient survey results are available in the online supplementary file.

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
