## [Reviewer comments · BMJ Open]

ARTICLE DETAILS

TITLE (PROVISIONAL)	Implementing online consultations in primary care: A mixed method evaluation extending normalisation process theory through service co-production
AUTHORS	Farr, Michelle; Banks, Jonathan; Edwards, Hannah; Northstone, Kate; Bernard, Elly; Salisbury, Chris; Horwood, Jeremy

VERSION 1 – REVIEW

REVIEWER	Glenn Robert King's College London, United Kingdom
REVIEW RETURNED	23-Oct-2017

GENERAL COMMENTS	Thank you for the opportunity to review this interesting paper. I have four main comments. The first two points below are relatively minor; the third requires more attention in my view. The final point is a methodological query which it would be helpful to have clarified. The specific study objectives could be more clearly and succinctly stated in the abstract. In the manuscript conclusions the authors state that 'E-consultations can increase patient access and satisfaction, but in their current form, were not perceived as creating sufficient workload efficiencies for continued practice usage'. The abstract rather weakens this conclusion by stating 'e-consultations were not an immediate solution for efficiency savings'. Given that (1) GPs felt their workload was being duplicated in many cases and (2) strikingly none of the 36 practices chose to adopt the technology after the pilot perhaps the concluding statement currently in the abstract warrants strengthening? I have two suggestions regarding the current rather brief discussion section. (1) Whilst the manuscript title suggests one contribution is the extension of 'normalisation process theory through service co-production' this issue is not returned to in the discussion section of the manuscript. This felt like an omission given the prominence in both the title and methods section - what are the authors reflections on the contribution made by their combination of NPT and service coproduction theory? Is it really an 'extension' of NPT? In what way has NPT been extended? (2) And related to this, does the use of 'touchpoints' as a lens to analyse some of the empirical data merit being equated with the application of 'service coproduction theory'?
---

	Given current debates about the nature and meanings of 'coproduction' (e.g. voluntary/involuntary) I felt the authors could provide a wider discussion in both these regards based on their empirical findings. Finally, it was not entirely clear to me how the 'touchpoints' were identified? What data source(s) were used and how were these analysed?
--	--

REVIEWER	Dr Clare Liddy Dept of Family Medicine, Bruyere Research Institute, University of Ottawa, Canada
REVIEW RETURNED	02-Nov-2017

GENERAL COMMENTS	Overall this is an interesting article that examines a very relevant topic- that of eConsultation from patient to provider. It does examine an area that is not well described in the literature yet and thus is an important study. I can see that the study was extensive and overall has merit. The qualitative approach was thorough and thoughtful. However I had a lot of difficulty getting through the paper as is. The first difficulty I had with the paper as written is lack of clarity about the actual research question- you talk about " an analysis of the implementation and acceptability ...to understand patient and provider experiences." This is very broad... There are then three data sets that are presented- both quantitative and qualitative (thus mixed methods) however not sure where the mixing happened and how that is then reflected in the results beyond the application of the NPT to mix survey and interviews but the quantitative piece was not well integrated. I found myself really going back and forth in the paper to try to put it all together . I was not sure why the quantitative data was presented as 'supplementary table" . I wonder if the authors have tried to put too much data in here...how was the quantitative data (case types, numbers,) used to inform the qualitative? The results as presented seem to mainly focus on qualitative - I think a table 1 demographics of the participants would be helpful. I would also like a brief summary of the quantitative data here- to give the reader a sense of how this was used, uptake, number of cases etc...then dive into the provider and patient perspectives... In summary, a relevant paper but needs reorganization and better focus to convey the study and main results/ key messages to the reader.
--

REVIEWER	Tracy Finch Northumbria University, United Kingdom
REVIEW RETURNED	06-Nov-2017

GENERAL COMMENTS	This is a well written paper, addressing the question of how to improve the implementation of online consultations in primary care. This paper draws together data from different methodological sources (staff interviews, patient records, patient survey responses), allowing for combined analysis of findings from different perspectives, and thus offers a novel approach to understanding the implementation of online consultations in general practice. There are a few points for the authors to consider:
---

	1. The abstract makes reference to the use of statistical analysis (chi square, odds ratios etc) but such analyses are not reported in the paper. For the objectives of this paper, descriptive use of quantitative data is appropriate but the reference in the abstract is misleading. 2. Can more information about the pilot service, and how it was developed? (and what role other stakeholders, such as patients, may have had in this). It is acknowledged that the paper is focusing on co-production in implementation (of core services) rather than service design, but some additional background on the service would help the interpretation of the findings presented later. 3. In using a mixed methods approach to evaluation, a challenge is in integrating data from different sources and acknowledging the strengths and limitations of what each data source brings to the integrative analysis. In this paper, staff perspectives are represented through qualitative interviews, but patient perspectives have been captured through open ended (free text) survey responses, that are likely to differ in depth (though perhaps offer some breadth of response, given the volume of participants). To help the reader to judge the relative contributions of the data sources, more information, particularly about the free text responses, could be provided – eg, although number of free text comments for the different survey questions is reported, can information about the length of comments (range, average) be provided to give the reader a feel for what may be drawn from this dataset? 4. In general, uptake was low, and of patients opting for online consultation, only 10% took part in the survey – although briefly acknowledged, the implications of this for the analysis presented in the paper should be more adequately discussed (eg – collection of patient survey data 7 days after their consultation, which for many will have been before a response from the service was received back, is a limiting factor here). This relates partly to the point above about comparative merits of the data sources. 5. Normalization Process Theory (NPT) is a useful approach for understanding problems of implementation, and has been widely used, especially in relation to electronic health/telehealthcare interventions. Some explicit comparison of the findings to previous studies of e-health using NPT would help put the findings in context, and allow more assessment of the utility of the framework – eg review of ehealth implementation structured by Mair et al, Bulletin of WHO, 2012 (http://cdrwww.who.int/bulletin/volumes/90/5/11-099424.pdf). There is an existing body of literature over the last 15 years that reflects the problems reported for online general practice consultation in this current paper, which is both disappointing (in terms of lack of progress in service design and improvement during this time) but important in that barriers to improving efficiencies in service provision using e-health technology are still not being addressed. 6. The authors have chosen to draw on co-production theory to highlight the interaction between staff and patients in the process of the online consultation. I would agree that NPT is a suitable vehicle for exploring extension through a co-production perspective, and yes, one critique of NPT currently is that is more professional/implementer focused in its lens. The foundations of NPT (focus on the ‘work’ of implementation, and how it is achieved collectively) are well aligned with a co-production philosophy and where the work of a change in service impacts on the active role of patients (as is the case here), they are key co-participants in the process.
--	---

	I feel that, given some of the constraints of the patient survey data, while bringing together co-production and NPT is a positive step, the analysis is more at a descriptive level at present. Some further discussion or elaboration of the role of a co-production emphasis – and distinguishing between ‘co-implementation’ and ‘co-production’ (the latter being more about developing together the best way of designing and delivering a new service) – would strengthen the analysis presented in the paper.
--	--

VERSION 1 – AUTHOR RESPONSE

REVIEWER 1

Comment: Thank you for the opportunity to review this interesting paper. I have four main comments. The first two points below are relatively minor; the third requires more attention in my view. The final point is a methodological query which it would be helpful to have clarified.

Response: Many thanks for your comments, questions and helpful suggestions. We have developed the paper in line with these, as follows below.

Comment: The specific study objectives could be more clearly and succinctly stated in the abstract.

Response: Objectives in the abstract (p.2) changed to read:

Objectives: To examine patient and staff views, experiences and acceptability of a UK primary care online consultation system and ask how the system and its implementation may be improved.

Comment: In the manuscript conclusions the authors state that 'E-consultations can increase patient access and satisfaction, but in their current form, were not perceived as creating sufficient workload efficiencies for continued practice usage'. The abstract rather weakens this conclusion by stating 'e-consultations were not an immediate solution for efficiency savings'. Given that (1) GPs felt their workload was being duplicated in many cases and (2) strikingly none of the 36 practices chose to adopt the technology after the pilot perhaps the concluding statement currently in the abstract warrants strengthening?

Response: Abstract conclusion (p.2) now reads:

Overall, the e-consultation system studied could improve access for some patients, but in its current form, it was not perceived by practices as creating sufficient efficiencies to warrant financial investment. We illustrate how this e-consultation system and its implementation can be improved, through mapping the co-production of e-consultations through touchpoints.

Comment: I have two suggestions regarding the current rather brief discussion section. (1) Whilst the manuscript title suggests one contribution is the extension of 'normalisation process theory through service co- production' this issue is not returned to in the discussion section of the manuscript. This felt like an omission given the prominence in both the title and methods section - what are the authors reflections on the contribution made by their combination of NPT and service coproduction theory? Is it really an 'extension' of NPT? In what way has NPT been extended?

(2) And related to this, does the use of 'touchpoints' as a lens to analyse some of the empirical data merit being equated with the application of 'service coproduction theory'?

(3) Given current debates about the nature and meanings of 'coproduction' (e.g. voluntary/involuntary) I felt the authors could provide a wider discussion in both these regards based on their empirical findings.

Response: Thank you for this helpful comment which has facilitated a greater reflection on the theoretical groundings and contributions of the article. Taking these separate points in order of how they have been responded to within the main paper, the following edits have been made in the Methods section (point 2) and the Discussion section (points 1 and 3):

(2) Does the use of 'touchpoints' as a lens to analyse some of the empirical data merit being equated with the application of 'service coproduction theory'?

In the Methods section: 'Theoretical integration of patient and staff data using NPT and co-production theory', we have further illustrated how both NPT and co-production were integrated, and then applied to the data. A new Figure 2 has been included, which had previously been taken out from the original paper to comply with author guidelines on numbers of figures and tables. New text (p.7) reads: Service co-production theory and NPT were theoretically integrated to examine not only implementation from staff and patient's points of view, but also the processes and interactions between patients and staff when using the e-consultation system. Service co-production can be understood as a process where service quality is shaped by (a) people's initial expectations of a service (b) staff and service users' roles, interactions and experiences within a service, leading to (c) their resulting satisfaction or dissatisfaction.^{23 29 34} Understanding this process helps to analyse service users' roles as a co-producer of a service.^{26 35} NPT constructs ^{14 16} and service co-production processes²³ can be integrated together and used to analyse staff and patients' initial expectations, interactions with and experiences of e-consultations, and their subsequent perceptions resulting in satisfaction/ dissatisfaction (Figure 2).

Figure 2: Combining NPT framework with service co-production processes

(1) and (3) A new section has been added to the Discussion, (p.17) titled:

NPT and service co-production theory

Service co-production theory and touchpoints can extend NPT through focussing on how technologies change the service process and interactions between patients and staff. Whilst involving patients voluntarily in co-designing technology may improve e-health technology;^{17 40} this paper's contribution illustrates how service co-production theory can support the analysis of how patients co-implement technology through everyday service interactions, rather than voluntarily being involved in co-designing a service. Service co-production particularly extends the collective action aspects of NPT, exploring in-depth how both staff and patients operationalise and relate through a service system. Touchpoint analysis illustrates how patients and staff responded to digital prompts and interacted through the e-consultation process. This fills a research gap to specifically examine how e-health services affect clinical interactions with patients.^{44 46} It shows how e-health implementation may be reconfigured through staff and service user produced knowledge⁴⁴ to improve technology and its implementation. This may tackle barriers to technological adoption, such as understanding how technology impacts care delivery, relationships between care givers and receivers, the role of patients in implementation, and how to maintain and improve ongoing implementation.^{17 46}

An extra sentence has also been included in the Conclusions (p.18):

Extending NPT through service co-production theory and touchpoints enables an analytic focus on service processes and interactions between staff and patients, and how the e-consultation system affected these.

Comment: Finally, it was not entirely clear to me how the 'touchpoints' were identified? What data source(s) were used and how were these analysed?

Response: A short description (p.7) has now been included about how service blueprinting techniques were used to identify touchpoints. Originally we had described this as a 'process map', to avoid conceptual overload for the reader. However it is an important methodological point to add, as to how these touchpoints were identified. This also highlights the relevance of service blueprinting techniques to understand service co-production²⁵:

'Touchpoints' (points of contact and interaction through a service process) were identified by using service blueprint techniques to map the e-consultation process.^{25 37} Service blueprints are maps of service systems that illustrate service user and staff roles, actions and interactions, and can illustrate how service users expectations and experiences affects service quality.^{25 37} Using staff interviews of the e-consultation process, and qualitative patient survey responses, three 'touchpoints'^{25 35} were identified, where patients and staff interacted through the e-consultation process.

REVIEWER 2

Comment: Overall this is an interesting article that examines a very relevant topic- that of eConsultation from patient to provider. It does examine an area that is not well described in the literature yet and thus is an important study. I can see that the study was extensive and overall has merit. The qualitative approach was thorough and thoughtful. However I had a lot of difficulty getting through the paper as is.

Response: Thank you for these positive comments. We hope we have suitably addressed the difficulties that you had with the paper, details as follows.

Comment: The first difficulty I had with the paper as written is lack of clarity about the actual research question- you talk about " an analysis of the implementation and acceptability ...to understand patient and provider experiences." This is very broad...

Response: Objectives in the abstract (p.2) changed to read:

To examine patient and staff views, experiences and acceptability of a UK primary care online consultation system and ask how the system and its implementation may be improved.

The research question at the end of the Background section (p.4) now reads:

This article analyses the implementation and acceptability of the eConsult system from patient and staff perspectives, using normalisation process theory (NPT)¹⁴ and service co-production theory²²⁻²⁹ to understand their experiences and how the e-consultation system and its implementation may be improved.

Comment: There are then three data sets that are presented-both quantitative and qualitative (thus mixed methods) however not sure where the mixing happened and how that is then reflected in the results beyond the application of the NPT to mix survey and interviews but the quantitative piece was not well integrated.

...how was the quantitative data (case types, numbers,) used to inform the qualitative?

Response: At what stage were methods 'mixed' and how is this reflected in the results?

Quantitative and qualitative researchers were a team that met regularly to discuss the research progress and emerging results. The quantitative and qualitative data collection occurred at the same time, so it was not the case that one data set then informed or followed the other. Instead data were combined at the analysis and write up stages of the research. Further details of how the different data sets have been integrated have been provided as follows.

1. The nature of the different data sets has been more explicitly stated at the beginning of the Methods section under the subtitle Research Design (p.4), explaining which data sets were quantitative and which were qualitative.

2. These different data sets were integrated at a number of different stages:

a. Under the heading 'Patient survey data' (p.7), it has been added that: Qualitative patient survey data was used to facilitate interpretation of the quantitative patient survey responses.

b. Under the heading 'Theoretical integration of patient and staff data using NPT and co-production theory' (p.7) it has been explained how:

Patient survey data (quantitative and qualitative), staff interview data and patient record data were theoretically integrated,³⁶ bringing different findings together into this theoretically-informed framework (Figure 2).

Please see response R1.3 that explains why this Figure has been added to the paper.

c. It has been explained how patient record data has been integrated with the analysis of staff and patient satisfaction, as follows (p.8):

Patients' clinical reasons for using an e-consultation and practice staff responses from patient record data (Table 2) were integrated with the analysis of qualitative staff and patients' comments about their satisfaction with the system, integrating all data sets. This integration of qualitative and quantitative data used established 'following a thread'³⁶ techniques where the question of why staff and patients were satisfied/ dissatisfied with the system, was traced using all data sets, to understand patients and staff sources of satisfaction/dissatisfaction with the system.

d. To further integrate data sets, an additional qualitative quote has been added to supplement quantitative data (p.12):

Where e-consultations resulted in a primary response of a prescription (7.2%), a 'fit note' statement of fitness for work (3.1%), test or treatment request (1.6%), referral (1.6%) or advice (9.1%) (occurring in 23% of patient e-consultation records),³⁰ these could save GP time as administrative staff relayed messages and there was no direct contact between the patient and GP. ADDED QUOTE

It helps in terms of administratively if there are things which can be done very simply, and that can free up, that can free up surgery time, to a degree (GP18).

Comment: I found myself really going back and forth in the paper to try to put it all together . I was not sure why the quantitative data was presented as 'supplementary table" . I wonder if the authors have tried to put too much data in here

Response: The quantitative data table A (Reason for e-consultation by primary response from practice staff) referred to was originally put in the supplementary file to keep with author guidance on the suggested numbers of tables and figures. However, we agree that this approach has made the paper become more qualitatively oriented.

We have taken this table A from the supplementary file and put it as Table 2 under Collective Action Touchpoint 3 (p.11-12) as this data fits best with the analysis on how clinicians interacted with patients following their e-consultation.

Additional text (p.11) has been added as follows:

Table 2 cross tabulates the primary clinical reason for patients using an e-consultation with the primary response to the e-consultation from practice staff. There were differences in GP responses according to patients' health queries. For example, medication queries and advice resulted in no face to face appointments, whilst neurological queries resulted in face to face appointments in 54% of cases.

In relation to the query about whether too much data has been put in this paper, we have two separate papers outlining the staff qualitative interviews of this study, and wider quantitative results on e-consultation usage. This paper under review integrates these published papers' findings with patient survey data through a theoretically informed integration.

Comment: The results as presented seem to mainly focus on qualitative - I think a table 1 demographics of the participants would be helpful. I would also like a brief summary of the quantitative data here- to give the reader a sense of how this was used, uptake, number of cases etc...then dive into the provider and patient perspectives...

Response: Steps have been made to adjust this and respond to the issues raised:

1. Table 1 in the Methods section already gives interview participant demographics.
2. Patient demographics and their reasons for consulting are presented in our separate BMJ Open quantitative paper on usage.

We have added a short description of these usage statistics and referenced this other paper that provides further details, under the results section Collective Action (p.9-10). This fits with the approach of theoretical integration of mixed methods³⁶ taken within the paper.

Over the 15 month pilot period, 7,472 patients completed an 'e-consultation', most frequently on weekdays and during traditional working hours.³⁰ Patient record data shows that women used e-consultations more than men (64.7% versus 35.3%) and 53.4% were between 25-44 years old. ³⁰ Most commonly, patients submitted administrative requests e.g. repeat prescriptions, test results and letters (22.5%), followed by immunological/ infection issues (14.4%) (see Table 2 and Edwards et al.³⁰).

3. Table 2 has been placed not at the beginning of the results section, but under Collective Action Touchpoint 3 (p.11-12) as this data fits best with the analysis on how clinicians interacted with patients following their e-consultation, following NPT themes.

Other associated edits with this comment include more clearly labelling theoretically informed illustrations as Figures, and data presentations as Tables.

Comment: In summary, a relevant paper but needs reorganization and better focus to convey the study and main results/ key messages to the reader.

Response: Edits made through the document have attempted to better convey the main results and key messages to the reader.

REVIEWER 3

Comment: This is a well written paper, addressing the question of how to improve the implementation of online consultations in primary care. This paper draws together data from different methodological sources (staff interviews, patient records, patient survey responses), allowing for combined analysis of findings from different perspectives, and thus offers a novel approach to understanding the implementation of online consultations in general practice.

There are a few points for the authors to consider:

Response: Thank you for your positive and encouraging comments on the paper. We hope that we have addressed the points that have been raised as follows.

Comment: 1. The abstract makes reference to the use of statistical analysis (chi square, odds ratios etc) but such analyses are not reported in the paper. For the objectives of this paper, descriptive use of quantitative data is appropriate but the reference in the abstract is misleading.

Response: Many thanks for pointing this out. Our other quantitative based paper in BMJ Open uses these statistical tests, but our data here is just based on descriptive statistics so the abstract (p.2) has been changed to ensure that it is appropriate to the data analysis used:
Descriptive statistics were used to analyse quantitative data.

Comment: 2. Can more information about the pilot service, and how it was developed? (and what role other stakeholders, such as patients, may have had in this). It is acknowledged that the paper is focusing on co-production in implementation (of core services) rather than service design, but some additional background on the service would help the interpretation of the findings presented later.

Response: A sentence has been included to explain how the eConsult system was developed (p.3), illustrating that patients weren't initially involved in co-designing the system:
The eConsult system was designed by GPs, software programmers and operational managers, with support from medical defence organisations.¹³

Whilst the above description illustrates that patients were not involved in its original design; the results of our study and the patient survey comments have informed the ongoing improvement of the system. A new reference⁴¹ has been added that illustrates this.

Reanalysis of interviews illustrates that:

Few practices reported involving patients in implementing e-consultations, one practice mentioned their patient participation group were concerned the system may disadvantage those who were less able to use technology. (Sentence added to Cognitive participation section p.9).

OneCare's own patient reference group, made up from representatives from Patient Participation Groups within GP practices, did discuss the e-consultation system pilot at a number of meetings, acting as a conduit to OneCare on what their practice's patients thought about eConsult. However, we can't demonstrate that this affected practice implementation processes, so we have not included this in the paper.

Comment: 3. In using a mixed methods approach to evaluation, a challenge is in integrating data from different sources and acknowledging the strengths and limitations of what each data source brings to the integrative analysis. In this paper, staff perspectives are represented through qualitative interviews, but patient perspectives have been captured through open ended (free text) survey responses, that are likely to differ in depth (though perhaps offer some breadth of response, given the volume of participants). To help the reader to judge the relative contributions of the data sources, more information, particularly about the free text responses, could be provided – eg, although number of free text comments for the different survey questions is reported, can information about the length of comments (range, average) be provided to give the reader a feel for what may be drawn from this dataset?

Response: Please see comments from reviewer 2 and responses (R2.2) above that provide further details of how the different data sets were analysed and integrated.

Please see R3.4 below for additional limitations of the data discussed. The details suggested about patient survey comments have now been added to Table B in the supplementary file, including range and average. Under Strengths and limitations (p.18) the following has been added:

Patients' qualitative survey comments varied in depth, but provided a wide breadth of responses, e.g. 510 respondents explained reasons for satisfaction/ dissatisfaction with the system. An early internal research report shared with the e-consultation software developers, has supported improvements to the e-consultation system studied.⁴¹

Comment: 4. In general, uptake was low, and of patients opting for online consultation, only 10% took part in the survey – although briefly acknowledged, the implications of this for the analysis presented in the paper should be more adequately discussed (eg – collection of patient survey data 7 days after their consultation, which for many will have been before a response from the service was received back, is a limiting factor here). This relates partly to the point above about comparative merits of the data sources.

Response: Included under Strengths and limitations heading in the discussion (p.18):

Patient surveys were only sent to patients who had submitted an e-consultation (of which 10% responded), thus representing a self-selecting sample of those who had invested time into the system. Surveys were sent to patients seven days after they had submitted an e-consultation, which may have been before their e-consultations had been processed with 14% of patients waiting to hear back.

Comment: 5. Normalization Process Theory (NPT) is a useful approach for understanding problems of implementation, and has been widely used, especially in relation to electronic health/telehealthcare interventions.

Some explicit comparison of the findings to previous studies of e-health using NPT would help put the findings in context, and allow more assessment of the utility of the framework – eg review of ehealth implementation structured by Mair et al, Bulletin of WHO, 2012 (<http://cdrwww.who.int/bulletin/volumes/90/5/11-099424.pdf>). There is an existing body of literature over the last 15 years that reflects the problems reported for online general practice consultation in this current paper, which is both disappointing (in terms of lack of progress in service design and improvement during this time) but important in that barriers to improving efficiencies in service provision using e-health technology are still not being addressed.

Response: Included new sentences in Key findings (p.16-17) to illustrate how our findings compare with other e-health literature and identified barriers to e-health implementation:

Other interventions designed to improve efficiency and access in primary care highlight potential workload issues; e.g. nurse-led telephone triage may reduce GP contact time, but increase overall clinician contact time.⁴³ Previous e-health studies that use NPT highlight barriers of adverse effects on workload⁴⁴ and poor interactional workability of technology which can impede adoption within primary care.⁴⁵ E-consultations supported efficiencies for straightforward GP queries, but less so complex ones, showing that how patients use technology can affect its implementation.⁴⁶ Our results align with other studies that highlight potential barriers to technological implementation including that: the clinical data the system was designed to generate from patients was sometimes incomplete;⁴⁷ the system was not fully interoperable with other IT systems, and costs prohibited long-term usage.¹⁷ Also included new paragraph in the discussion section (p.17), following Reviewer 1's comments (R1.3), 'NPT and service co-production theory'. This includes a discussion of the utility of the new framework, and how it may fill current research gap:

This fills a research gap to specifically examine how e-health services affect clinical interactions with patients.⁴⁴ ⁴⁶ It shows how e-health implementation may be reconfigured through staff and service user produced knowledge⁴⁴ to improve technology and its implementation. This may tackle barriers to technological adoption, such as understanding how technology impacts care delivery, relationships between care givers and receivers, the role of patients in implementation, and how to maintain and improve ongoing implementation.¹⁷ ⁴⁶

Finally new material has been added to the Policy and practice implications (p.17), building on ongoing debates and developments within the implementation of online consultation systems: NHS England has offered financial support for practices to adopt online consultations.⁶ Our research affirms that clear implementation guidance is needed⁴⁹ and provides recommendations to support the technological developments of e-consultations and future implementation to alleviate additional GP workload whilst improving patient access. NHS England case studies of e-consultation systems include their potential role to triage most patients.⁵⁰ ⁵¹ Whilst our study gave no statistical evidence that patient socioeconomic factors affected usage rates,³⁰ practitioners in our qualitative study had concerns about the system's potential impact on equality of access. Further research is needed to investigate equity of access when implementing e-consultations.

To provide data for these issues we have add a short section under Reflexive Monitoring (p.14):

Some practice staff were also concerned that the system might exacerbate inequalities of access for people with literacy difficulties or whose first language is not English, and those with difficulties in using a computer or mobile device.

Comment: 6. The authors have chosen to draw on co-production theory to highlight the interaction between staff and patients in the process of the online consultation. I would agree that NPT is a suitable vehicle for exploring extension through a co-production perspective, and yes, one critique of NPT currently is that is more professional/implementer focused in its lens. The foundations of NPT (focus on the 'work' of implementation, and how it is achieved collectively) are well aligned with a co-production philosophy and where the work of a change in service impacts on the active role of patients (as is the case here), they are key co-participants in the process.

I feel that, given some of the constraints of the patient survey data, while bringing together co-production and NPT is a positive step, the analysis is more at a descriptive level at present. Some further discussion or elaboration of the role of a co-production emphasis – and distinguishing between ‘co-implementation’ and ‘co-production’ (the latter being more about developing together the best way of designing and delivering a new service) – would strengthen the analysis presented in the paper.

Response: Please see responses to Reviewer 1 (R1.3). Two main areas of the paper have been developed with respect to this comment:

1. Further explanation of how NPT and service co-production theory can be integrated together has now been added within the Methods section: ‘Theoretical integration of patient and staff data using NPT and co-production theory’ to illustrate how both theories are applied within the data. A new Figure 2 has been included, (p.7) which had been omitted from the original paper to comply with author guidelines on numbers of figures and tables.

Service co-production theory and NPT were theoretically integrated to examine not only implementation from staff and patient’s points of view, but also the processes and interactions between patients and staff when using the e-consultation system. Service co-production can be understood as a process where service quality is shaped by (a) people’s initial expectations of a service (b) staff and service users’ roles, interactions and experiences within a service, leading to (c) their resulting satisfaction or dissatisfaction.^{23 29 34} Understanding this process helps to analyse service users’ roles as a co-producer of a service.^{26 35} NPT constructs 14 16 and service co-production processes²³ can be integrated together and used to analyse staff and patients’ initial expectations, interactions with and experiences of e-consultations, and their subsequent perceptions resulting in satisfaction/ dissatisfaction (Figure 2).

2. A new section has been added to the Discussion, (p.17) titled:

NPT and service co-production theory

Service co-production theory and touchpoints can extend NPT through focussing on how technologies change the service process and interactions between patients and staff. Whilst involving patients voluntarily in co-designing technology may improve e-health technology;^{17 40} this paper’s contribution illustrates how service co-production theory can support the analysis of how patients co-implement technology through everyday service interactions, rather than voluntarily being involved in co-designing a service. Service co-production particularly extends the collective action aspects of NPT, exploring in-depth how both staff and patients operationalise and relate through a service system. Touchpoint analysis illustrates how patients and staff responded to digital prompts and interacted through the e-consultation process.

OTHER UPDATES

In addition to these reviewers comments, we have made the following additions to the paper, to illustrate how our research evidence can inform current debates on e-consultations. Since our original submission, NHS England have launched the £45 million fund for e-consultations, including new case studies. There have also been new debates on our recently released BJGP paper from this study in BMJ with responses from software developers.

Response: 1. Ellender’s response to our research has been referenced⁴¹, to illustrate how the e-consultation system has developed as a result of our research.

2. Sentences have been added in relation to new NHS England case studies that promote the release of the e-consultation fund, under subtitle: Policy and practice implications, illustrating where further research is needed (see R3.5).

NHS England case studies of e-consultation systems include their potential role to triage most patients.^{50 51} Whilst our study gave no statistical evidence that patient socioeconomic factors affected usage rates,³⁰ practitioners in our qualitative study had concerns about the system’s potential impact on equality of access. Further research is needed to investigate equity of access when implementing e-consultations.

3. Some changes have also been made to Table 1 to ensure that the practice profiles that we give here do not make the practices identifiable.

VERSION 2 – REVIEW

REVIEWER	Glenn Robert King's College London, United Kingdom
REVIEW RETURNED	01-Jan-2018

GENERAL COMMENTS	Thank you for your clear and extensive revisions. I have no further comments.
---

REVIEWER	Clare Liddy University of Ottawa
REVIEW RETURNED	11-Jan-2018

GENERAL COMMENTS	I think the authors have been very thorough in responding to my previous concerns. Thank you
---

REVIEWER	Tracy Finch Northumbria University, United Kingdom
REVIEW RETURNED	19-Dec-2017

GENERAL COMMENTS	The authors have made substantial revisions in response to the collective reviewers' comments, and the manuscript is now significantly improved as a result.
--